Transcriptomic analysis of melon/squash graft junction reveals molecular mechanisms potentially underlying the graft union development

Xu Chuanqiang chuanqiang79@syau.edu.cn 1 2 3
Zhang Ying 1 2 3
Zhao Mingzhe 4
Liu Yiling 1 2 3
Xu Xin 1 2 3
Li Tianlai ltl@syau.edu.cn 1 2 3
1 College of Horticulture, Shenyang Agricultural University , Shenyang , Liaoning , China
2 Collaborative Innovation Center of Protected Vegetable Surround Bohai Gulf Region , Shenyang , Liaoning , China
3 Key Laboratory of Protected Horticulture (Shenyang Agricultural University) Ministry of Education , Shenyang , Liaoning , China
4 College of Agronomy, Shenyang Agricultural University , Shenyang City , Liaoning Province , China
Gibson Susan
Electronic publication date: 2021 Dec 13
Publication date: 2021
Volume: 9
Electronic Location ID: e12569
Received 2020 Aug 11; Accepted 2021 Nov 8
Copyright: ©2021 Xu et al.
Copyright year: 2021
Copyright holder: Xu et al.
License: This is an open access article distributed under the terms of the Creative Commons Attribution License, which permits unrestricted use, distribution, reproduction and adaptation in any medium and for any purpose provided that it is properly attributed. For attribution, the original author(s), title, publication source (PeerJ) and either DOI or URL of the article must be cited.
License URL: https://creativecommons.org/licenses/by/4.0/

Keywords: Hormone, Graft healing, Lignin, Oriental melon, Squash, Transcriptome

Funding: The National Natural Science Foundation of China 31401917 Basic Research Project of Liaoning Province LSNJC202005 Cultivation Plan for Youth Agricultural Science and Technology Innovative Talents of Liaoning Province 2014050 China Agriculture Research System CARS-25 This project was funded by the National Natural Science Foundation of China (Grant No. 31401917) and the Basic Research Project of Liaoning Province (Grant No. LSNJC202005), the Cultivation Plan for Youth Agricultural Science and Technology Innovative Talents of Liaoning Province (Grant No. 2014050), and the China Agriculture Research System (CARS-25). The funders had no role in study design, data collection and analysis, decision to publish, or preparation of the manuscript.

==============================
Oriental melon (Cucumis melo var. makuwa Makino) has become a widely planted horticultural crop in China especially in recent years and has been subjected to the grafting technique for the improvement of cultivation and stress resistance. Although grafting has a long history in horticulture, there is little known about the molecular mechanisms of the graft healing process in oriental melon. This study aims to reveal the molecular changes involved in the graft healing process. In the present work, anatomical observations indicated that the 2, 6, and 9 DAG were three critical stages for the graft healing and therefore, were selected for the subsequent high-throughput RNA-seq analysis. A total of 1,950 and 1,313 DEGs were identified by comparing IL vs. CA and CA vs. VB libraries, respectively. More DEGs in the melon scion exhibited abundant transcriptional changes compared to the squash rootstock, providing increased metabolic activity and thus more material basis for the graft healing formation in the scion. Several DEGs were enriched in the plant hormone signal transduction pathway, phenylpropanoid biosynthesis, and carbon metabolism. In addition, the results showed that concentrations of IAA, GA3, and ZR were induced in the graft junctions. In conclusion, our study determined that genes involved in the hormone-signaling pathway and lignin biosynthesis played the essential roles during graft healing. These findings expand our current understandings of the molecular basis of the graft junction formation and facilitate the improvement and success of melon grafting in future production.

Introduction

Oriental melon (Cucumis melo var. makuwa Makino) is a high-value horticultural crop. However, during the actual production of melon, problems such as obstacles from continuous cropping and soil-borne diseases have caused yield loss and reduced fruit quality. Plant grafting is a biological phenomenon that involves the physical connection of two plants to produce a chimeric organism (Melnyk et al., 2015). A common practice and useful tool across the world is to graft susceptible vegetable scions onto resistant rootstocks to increase tolerance to soil-borne diseases and environmental stresses, therefore improving yields. In China, plantation of grafted-melons plants contributes to approximately more than 90% of the total melon protected cultivation area. Despite the wide-spread application of grafting in horticultural production and scientific research, biological mechanisms of plant grafting remain poorly understood.

The graft healing is the process by which an integrated grafted plant forms through the interaction of different tissues, organs, and cells. Although the length of time required for the healing process can vary with different varieties, ages, and grafting methods, the healing process steps are identical (Roberts, 1949). Since the graft healing is a complex process of tissue regeneration, it can be divided into three key events: (1) tissue adhesion between the scion and the rootstock and the formation of the barrier layer, (2) callus formation, and (3) reconnection of the vascular bundle bridge between the scion and the rootstock (Flaishman et al., 2008; Jeffree & Yeoman, 1983; Moore & Walker, 1981). Moreover, many reports suggest that plant hormones and antioxidants play essential roles in promoting the graft healing development (Aloni et al., 2010; Asahina et al., 2002; Fernandez-Garcia, Carvajal & Olmos, 2004; Melnyk & Meyerowitz, 2015). Understanding the healing mechanism of plant grafting is of considerable significance to improving the survival rate of grafted seedlings and the application of the grafting technique in production.

Recent reports on graft healing have focused on the analysis of histological, physiological, and biochemical changes from developmental perspectives. However, our understanding of the molecular mechanisms of successful grafting is insufficient (Chen et al., 2017; Cookson et al., 2013). Previous studies have found that transcriptomic data can be used to analyze wound healing and tissue reunion (Cheong et al., 2002). In grapevine, transcriptome analysis has revealed many genes associated with secondary metabolism, cell wall modification, signaling, and wound responses (Cookson et al., 2013). In hickory, cDNA-AFLP technology was used to detect gene expression at different time-points during the graft healing process. From this study, 49 DEGs related to the amino acid metabolism, auxin transport, signal transduction, water metabolism, cell cycle, and substance secretion were identified, confirming that graft healing is a complex metabolic process (Zheng et al., 2009). In another study on the graft healing process in hickory, transcriptomic analysis has revealed the differential expression of 112 candidate unigenes, which are involved in the auxin and cytokinin signaling pathways (Qiu et al., 2016). In watermelon, transcriptomic information has provided new insights into different biological and metabolic processes involved in response to grafting (Liu et al., 2015). In Litchi species, nine DEGs annotated in the auxin signaling pathway were up-regulated when compared with incompatible grafts from transcriptomic analysis of compatible graft healing (Chen et al., 2017). Yin et al. (2012) investigated transcriptomes of Arabidopsis thaliana grafts, and the results revealed that a large number of IAA-related genes were elevated within one day after grafting. Other studies suggest that the inter-tissue communication would act as an activating signal to vascular regeneration, which occurs independently of functional vascular connections (Melnyk et al., 2018). These studies show that transcriptome analysis could be a valuable approach for investigating molecular mechanisms underlying grafting-specific biochemical processes. The application of RNA-seq has accelerated gene-expression profiling and gene identification specific to grafting in many plant species.

Since grafted melons are currently widely used in melon production and molecular mechanisms critical to graft-healing are still mostly elusive, we aimed to identify new grafting-response molecular mechanisms at the transcriptional level. We did this by sequencing the transcriptome of oriental melon grafts and characterized the transcriptomic changes at critical points across the graft healing process.

Materials and Methods

Plant materials and grafting procedures

Scion: Oriental melon (Cucumis melo var. makuwa.), YinQuan No. 1 cultivar, was obtained from the Vegetable Research Institute, Liaoning Academy of Agricultural Science, Shenyang, China.

Rootstock: Squash (C. moschata), ShengZhen No. 1 cultivar, was obtained from SHENGDIYA Agricultural High-Tech Company (Shenyang, China).

The splicing method was used for grafting. Grafting was carried out when the scion’s first-true-leaf fully expanded and rootstock’s cotyledon development stage, using the one-cotyledon method (Davis et al., 2008). Grafted seedlings were transplanted in the nutritional bowl (12 cm × 12 cm) in a greenhouse at Shenyang Agriculture University, Liaoning, China (41°49′N 123°32′E). To maintain a high humidity and facilitate graft formation, grafted seedlings were moved into the healing chamber with 26 ± 3 °C temperature and 90–95% relative humidity condition. After 72 h, the grafted seedlings were gradually exposed to daylight and lower relative humidity (about 60%) by adjusting the humidifier and shading level over 5 days. The healed plants were removed from the healing chamber 8 days after grafting and grown in the greenhouse (24 ± 3 °C) in a shaded area for 2 days before full exposure to sunlight (Liu et al., 2017).

Paraffin sectioning and microscopy

We sampled 0.3–0.5 cm stems above and below the graft junction. The samples (2–9 d) were fixed, softened, dehydrated, infiltrated, and embedded in paraffin as described by Ribeiro et al. (2015). Transverse serial sections approximately 10 µm thick were cut and stained with pH4.4 toluidine blue (O’Brien, Feder & McCully, 1964, modified), and mounted using synthetic resin (Permount). Sections were examined using a light microscope (Lecia RM 2245, Germany).

RNA extraction, library construction, and sequencing

Screening through paraffin sections, samples were taken from graft junctions at the IL stage (2 DAG), CA stage (6 DAG), and VB stage (9 DAG) with three biological replicates and about 0.6 g per sample. Quality of the total RNA extraction, library construction, and sequencing was verified by the Biomarker Technology Co. according to the standard operating protocols (Beijing, China, https://www.biocloud.net/). The concentration of extracted total RNA was measured using the NanoDrop 2000 (Thermo Fisher, USA). The integrity of RNA was assessed using an RNA Nano 6000 Assay Kit for the Agilent Bioanalyzer 2100 system (Agilent Technologies, Santa Clara, CA, USA).

A total of 1 µg RNA per sample was used as input material for the RNA sample preparation. Sequencing libraries were generated using a NEBNext UltraTM RNA Library Prep Kit for Illumina (NEB, Ipswich, MA, USA) following the manufacturer’s recommendations and index codes were added for attributing sequences to each sample. PCR was performed with a Phusion High-Fidelity DNA polymerase, Universal PCR primers, and Index (X) Primer, and the PCR products were purified (AMPure XP system) and library quality was assessed on the Agilent Bioanalyzer 2100 system (Guo et al., 2015). The sequencing raw data was deposited in the NCBI Sequence Read Archive (SRA) with the accession number PRJNA655799.

Differential expression analysis

We separately used Melon (DHL92) v3.6.1 Genome and Cucurbita moschata (Rifu) Genome (http://cucurbitgenomics.org) to analysis the sequencing raw data of graft junction. The differential expression analysis on two conditions/groups was performed using the DEseq. The DEseq provides statistical procedures for determining the differential expression in the digital gene-expression-data using a model based on the negative binomial distribution. Benjamini and Hochberg method for controlling the false discovery rate were used to adjust the resulting P-values. Genes with an adjusted P-value < 0.01 were considered to be differentially expressed (Zhang et al., 2018).

GO enrichment and KEGG enrichment

We used the GOseq R package based on Wallenius’ non-central hypergeometric distribution (Young et al., 2010) to analyze Gene Ontology (GO) enrichment of the DEGs, and KOBAS (Mao et al., 2005) software to test the statistical enrichment of DEGs in KEGG (http://www.genome.jp/kegg/) pathways.

The accuracy of RNA-seq data was validated using qRT-PCR with selected DEGs. qRT-PCR was performed with three biological replicates for each sample. The first-strand cDNA synthesis was performed using a Prime-Script™ II First Strand cDNA synthesis kit (Takara Bio, Dalian, China) according to the manufacturer’s instructions. The primer sets for each unigene were designed with Primer Premier 5.0 (Table S6). qRT-PCR reactions were carried out on a Yena Real-Time PCR System (qTOWER3/qTOWER3 touch, Germany) using SYBR Premix Ex Taq™ II kit (Takara, Japan). Transcriptional abundance was calculated using the 2−ΔΔCt method and normalized to the reference actin gene.

Detection of hormone content by ELISA

Samples were taken from graft junctions at the IL stage, CA stage, and VB stage with three biological replicates. The content of IAA, ZR, GA, and ABA was measured using the Enzyme-Linked Immunosorbent Assay (ELISA). ELISA kits for these hormones were developed by the China Agricultural University. The accuracy of ELISA kits was validated by GC-MS and HPLC methods. A detailed protocol for determining the hormone content was previously described in Yang et al. (2001).

Statistical analysis

The data are presented as the mean ± standard deviation of three replicate samples. A one-way analysis of variance (ANOVA) was performed using the SPSS 22.0. The significance was determined using ANOVA followed by Duncan’s multiple range tests for experiments at P < 0.05. The figures were generated by using Origin 8.0 software.

Results

Anatomical observation

To determine the vital stages of the healing process in oriental melon grafts, we continuously monitored histological changes in sections of the graft junction on the microscope for 8 days following grafting. The graft healing formation was divided into the following three recognizable developmental stages: isolated layer (IL) stage, callus (CA) stage, and vascular bundles (VB) stage (Fig. 1) (Moore, 1982). At the IL stage (2 DAG), dark-stained cell layers were observed on the surface of the wound, which should originate from cells of damaged tissues forming the isolated layer (Fig. 1A). Most previously published reports have shown that the production of this isolated layer results from the wounding response (Yeoman et al., 1978). The function of the isolated layer could be to reduce the loss of water and to resist the subsequent infection of pathogens. The isolated layer gradually disappeared with the increase in the length of time following grafting, and the color in the wound interface became lighter forming the callus at the CA stage (6 DAG) (Fig. 1B). The formation of callus tissue provides a communication bridge for the scion and the stock. However, the formation of a callus was recognized as a passive wounding response, which was not associated with the compatibility reaction (Wang & Kollmann, 1996). At the VB stage (9 DAG), new vascular tissue formed, connecting the xylem and the phloem between the scion and the stock (Fig. 1C). This connection of vascular tissues between grafting partners was a mark for successful grafting (Olmstead et al., 2006; Pina & Errea, 2005). Based on the above observations, we selected 2, 6, and 9 DAG, respectively named the IL stage, the CA stage and the VB stage as three vital time points for sample collection and subsequent transcriptome studies.

Transcriptome sequencing and analysis

Following sequence quality control, a total of 125.84 Gb clean data was obtained (Table S1). The clean reads of all samples were mapped to the reference genomes of melon and squash with efficiencies ranging from 9.57 to 27.90% and 60.19 to 76.82%, respectively. The sequence validation of transcriptomes was greater or equal to 30 (Q30), indicating that the sequencing data were of high quality and reliability.

Figure 1 Anatomical observation of graft junction tissue using paraffin sectioning and microscopy method underlying the graft union development.

(A) The IL stage, the isolated layer stage. (B) The CA stage, the callus stage. (C) The VB stage, the vascular stage bundles. SC, scion. RT, rootstock.

Figure 2 The number of differentially expressed genes of the IL stage vs. the CA stage and the CA stage vs. the VB stage.

(A, C) Number of up- and down-regulated DEGs in the genome of Melon (DHL92) v3. 6.1 and Cucurbita moschata (Rifu). (B, D) Distinct expression regulation of transcripts represented by a Venn Diagram. (A, B) oriental melon (Cucumis melo var. makuwa.); (C, D) squash (C. moschata).

Differentially expressed genes identification

Pairwise comparison of graft healing revealed DEGs (≥2-fold change and FDR < 0.01) at IL stage, CA stage, and VB stage. Two DEG libraries (IL stage vs. CA stage, CA stage vs. VB stage) were generated (Fig. 2). A total of 1,440 DEGs were identified in the melon genome and 510 DEGs in the squash genome when the IL stage was compared to the CA stage. In the CA stage vs. VB stage library, 1,043 melon and 270 squash DEGs were identified. Overall, more transcriptional changes were identified in the melon genome in comparison to the squash genome during the complete graft healing process. Furthermore, less differential transcription activity was observed as the graft healing progressed into later developmental stages. Within the IL stage vs. the CA stage melon DEGs, 749 were up-regulated and 691 were down-regulated, whereas 127 squash DEGs were up-regulated and 383 were down-regulated, showing a repression-biased shift for the squash transcriptome compared to a more even-balanced transcriptome for the melon scion during the earlier graft healing process. On the other hand, as represented in the CA stage vs. VB stage library, 689 up-regulated and 354 down-regulated melon DEGs were identified, suggesting an activation-biased shift in the transcriptome for the melon scion. Also, the identification of 166 up-regulated and 104 down-regulated squash DEGs suggested that squash stock reached a relatively balanced activation-repression status during the later graft healing process.

Taken together, much more dynamic changes in the transcriptional profile of the melon scion were observed than those in the squash rootstock, indicating that the melon scion may play a more significant role in the initiation of the graft healing process, contribute more vigorous metabolic activities, and therefore promote the healing process.

Figure 3 GO functional classification and enrichment analysis of differentially expressed genes of the IL stage vs. the CA stage using Melon (DHL92) v3.6.1 Genome.

Functional categorization

Analysis of the DEGs biological functions during the graft healing process in the melon scion was carried out using the Gene Ontology (GO). Most DEGs were annotated into three functional categories: biological process, cellular component, and molecular function (Figs. 3–6; Table S2). Within the biological process category, the most highly represented terms were ‘metabolic processes’, ‘single-organism processes’, and ‘cellular processes’. Within the cellular component category, ‘cell membranes’ and ‘cells’ were the two most abundant terms indicating biological activity in the cell and organelle membrane. Within the ‘molecular function’ category, ‘binding’, ‘catalytic activity’, and ‘transporter activity’ were highly enriched. There was no obvious difference in DEGs functional annotation between melon and squash. However, the number of DEGs separated in each functional category was different. These results showed that the DEGs could perform catalytically and transporting molecular functions, participate in metabolic biology activities, and provide a material basis for the graft healing process. Therefore, candidate genes related to these processes can potentially play crucial roles in graft healing formation.

Figure 4 GO functional classification and enrichment analysis of differentially expressed genes of the CA stage vs. the VB stage using Melon (DHL92) v3.6.1 Genome.

Figure 5 GO functional classification and enrichment analysis of differentially expressed genes of the IL stage vs. the CA stage using Cucurbita moschata (Rifu) Genome.

Figure 6 GO functional classification and enrichment analysis of differentially expressed genes of the CA stage vs. the VB stage using Cucurbita moschata (Rifu) Genome.

KEGG pathway analysis was then performed to characterize the main metabolic and signal transduction pathways enriched from the DEGs. The top twenty enriched pathways based on numbers and enrichment-levels are presented in Figs. 7–10 and Table S3. In the melon scion, DEGs involved in phenylpropanoid biosynthesis, plant hormone signal-transduction, biosynthesis of amino acid, and carbon metabolism were identified in the IL stage vs. CA stage (Fig. 7). On the other hand, linoleic synthesis and phenylpropanoid biosynthesis were highly enriched for the CA stage vs. VB stage (Fig. 8). In the squash rootstock, the phenylpropanoid biosynthesis pathway was found to be significantly enriched during the early graft healing process (Figs. 9, 10). Therefore, this was the one common KEGG metabolic pathway shared by the scion and the rootstock from both the IL stage vs. the CA stage and the CA stage vs. the VB stage libraries. Our study also showed that the metabolic pathways enriched in the scion should have more activity than in the rootstock. In short, phenylpropanoid biosynthesis and plant hormone signal transduction were the overlapping pathways identified by cross-examinations during the graft healing process. Therefore, we directed the focus of the study to these two vital metabolic pathways in the sequencing data.

Figure 7 Enrichment analysis of KEGG pathways of the IL stage vs. the CA stage using Melon (DHL92) v3.6.1 Genome.

Figure 8 Enrichment analysis of KEGG pathways of the CA stage vs. the VB stage using Melon (DHL92) v3.6.1 Genome.

Figure 9 Enrichment analysis of KEGG pathways of the IL stage vs. the CA stage using Cucurbita moschata (Rifu) Genome.

Figure 10 Enrichment analysis of KEGG pathways of the CA stage vs. the VB stage using Cucurbita moschata (Rifu) Genome.

Hormone signal-transduction pathways were activated during grafting

Plant hormones responsible for scion-rootstock communication are considered to be essential for graft healing formation (Melnyk et al., 2015). Our results identified a variety of genes involved in hormone signal-transduction pathways (Figs. 11–14). The transcriptional levels of most auxin transporter-encoding genes significantly changed during the graft healing process (Fig. 11). Unigenes encoding AUX1 (except MELO3C003035.2) and auxin/indole-3-acetic acid (AUX/IAA) were down-regulated at CA compared with IL, which were then up-regulated during the later stage of grafting. Similar AUX/IAA expression profiles were observed both in the melon scion and the squash rootstock. Furthermore, four GH3 genes (MELO3C027346.2, MELO3C007597.2, and MELO3C008672.2 in the melon scion and Cmoch20G005830 in the squash stock) showed a downward trend at the early stage. Accordingly, the content of IAA declined at the CA stage, when the graft healing process was completed, but the IAA content increased (Fig. 15A). These results indicate an IAA-response during wound healing.

Figure 11 Transcript abundance changes of auxin signaling-related genes of graft junction tissue at IL, CA, VB stage underlying the graft union development.

Figure 12 Transcript abundance changes of cytokinin signaling-related genes of graft junction tissue at IL, CA, VB stage underlying the graft union development.

Figure 13 Transcript abundance changes of gibberellin signaling-related genes of graft junction tissue at IL, CA, VB stage underlying the graft union development.

Figure 14 Transcript abundance changes of abscisic acid signaling-related genes of graft junction tissue at IL, CA, VB stage underlying the graft union development.

Figure 15 The IAA, ZR. ABA and GA content of graft junction tissue at the IL, CA, VB stage underlying the graft union development.

(A), IAA. (B), ZR. (C), ABA. (D), GA. Different letters over the bars denote significance at P < 0.05 by ANOVA.

Genes involved in the cytokinin signaling pathway were identified from the melon transcriptome data (Fig. 12). Consistent with its role in the graft healing formation, we found that ZR content was elevated at the CA stage and VB stage (Fig. 15B). Two unigenes encoding the AHP protein and one encoding a type-A ARR protein were significantly up-related at the early stage of grafting, suggesting their activation for the callus formation stage. These genes showed a down-regulated pattern during the late healing stage. In contrast, another unigene encoding CRE1 was up-regulated during the early stage, but then down-regulated at the VB stage. Two unigenes encoding type-B ARR were up-regulated at the late stage.

Most previous reports have suggested that ABA can be inhibitory to wound healing (Ikeuchi et al., 2017). Therefore, the content of ABA in the graft junction was measured by ELISA and results showed a significant decrease at the CA stage and VB stage (Fig. 15C). For genes involved in the ABA-signaling pathway (Fig. 14), only one PYR/PYL-encoding gene exhibited a high expression level at the VB stage compared with the CA stage. Four PP2C genes were up-regulated during the early grafting stages and then decreased during the late VB period.

Gibberellins have been identified as xylogenesis-triggering hormones (Mauriat & Moritz, 2009; Ragni et al., 2011), which are possibly vital for vascular formation. In our transcriptome data, a GID1 gibberellin receptor (MELO3CO15880.2), TF (MELO3CO26410), and GID2-encoding gene (CmoCh01G018640) were significantly up-regulated at the late stage. Furthermore, one TF (CmoCh09G007180) was induced at the early stage and then decreased at the VB stage (Fig. 13). The content of GA increased significantly at the VB stage but showed no significant difference at the IL stage and CA stage (Fig. 15D). Our results suggest that GAs may be involved in vascular formation.

In summary, most of the genes in the melon scion were enriched in the CTK-signaling transduction pathway to promote the callus formation and genes in the IAA and GA-signaling transduction pathways were induced to promote the vasculature formation. The results also implied the important role of scion in the healing process. The phylogenetic data of plant hormone signal transduction genes were shown in Table S4.

Lignin biosynthesis during graft healing

Lignin is the second most abundant macromolecular organic substance in plants. It carries out important biological functions and plays an essential role in the formation of vascular bundles (Zhong et al., 2000). Hence, further analysis of genes involved in lignin synthesis was performed. Over 15 DEGs were characterized as related to five of the enzymes in the lignin biosynthesis pathway including genes encoding phenylalanine ammonia-lyase (PAL), catechol-O-methyl transferase (COMT), 4-coumarate-coenzyme A ligase (4CL), caffeoyl-CoA O-methyltransferase (CCoAOMT), and cinnamyl-alcohol dehydrogenase (CAD). Most PAL genes were down-regulated significantly at the early stage and then increased at the VB stage. Other DEGs enriched in COMT, 4CL, CAD, and CCoAOMT also showed an increasing trend during the graft healing process (Fig. 16). The content of lignin was consistent with the expression level of most lignin DEGs (Fig. S1). The lignin biosynthesis-related genes in melon and squash libraries were up-regulated during the late stage, explaining the role of vasculature reconnection. The phylogenetic data of lignin biosynthesis genes are presented in Table S5.

Figure 16 Transcript abundance changes of lignin biosynthesis-related genes of graft junction tissue at IL, CA, VB stage underlying the graft union development.

Validation of RNA-seq Data by Real-time PCR

To verify the DEGs related to hormone signaling and lignin biosynthesis identified in the data, qRT-PCR assays were performed with samples independently collected from graft healing during different graft union development stages (Fig. 17). The expression levels of 12 selected genes including CmoCh07G009530 (Fig. 17A), CmoCh08G005650 (Fig. 17B), CmoCh08G003030 (Fig. 17C), CmoCh20G005830 (Fig. 17D), MELO3C034560.2 (Fig. 17E), MELO3C026019.2 (Fig. 17F), MELO3C010317.2 (Fig. 17G), MELO3C015359.2 (Fig. 17H), MELO3C014091.2 (Fig. 17I), MELO3C004382.2 (Fig. 17J), MELO3C007691.2 (Fig. 17K), MELO3C015880.2 (Fig. 17L) were essentially consistent with the RNA-Seq results. The primer sequences are listed in Table S6.

Figure 17 Differential gene verification of graft junction tissue at the IL, CA, VB stage underlying the graft union development.

(A–D) Cucurbita moschata (Rifu) Genome. (E–M) Melon (DHL92) v3.6.1 Genome. (A) CmoCh07G009530, (B) CmoCh08G005650, (C) CmoCh08G003030, (D) CmoCh20G005830, (E) MELO3C034560.2, (F) MELO3C026019.2, (G) MELO3C010317.2, (H) MELO3C015359.2, (I) MELO3C014091.2, (J) MELO3C004382.2, (K) MELO3C007691.2, (L) MELO3C015880.2. Different letters over the bars denote significance at P < 0.05 by ANOVA.

Discussion

Grafting is an effective method in oriental melon production adopted for overcoming soil-borne diseases and continuous-cropping obstacles. Therefore, graft healing is a critical but complicated developmental process during plant grafting. Understanding the mechanisms underlying graft healing is therefore of both theoretical and applicable importance. Recently, RNA-seq has become a rapidly adopted technology for studying the dynamics of transcriptomes, which can also be used in the exploration of transcriptional regulatory mechanisms during the grafting process (Chen et al., 2017; Cheong et al., 2002; Cookson et al., 2013; Liu et al., 2015; Melnyk et al., 2018; Mo et al., 2018). To our knowledge, this is the first study to use RNA-seq to analyze a large number of genes involved in the graft healing process in oriental melon grafts.

In the present study, a total of 2,483 and 780 genes were differentially expressed in the genome of melon scion and squash rootstock, respectively (Fig. 2). The transcriptomic changes in the melon scion were significantly more dynamic than those in the squash rootstock, suggesting that the melon scion is more resourceful in promoting the graft healing formation. Although the rootstocks affect the scions development (Gregory et al., 2013; Lee et al., 2010), the scions usually have a bigger effect on the rootstocks than the other ways round, it is just rarely studied, DEGs in grafted melon potentially reflected the regulatory functions during callus formation and vascular bundle development. Significantly more melon DEGs were found to be involved in these pathways in the scion compared to the squash rootstock. Functional categorization of identified DEGs suggested that the grafting healing process triggered the expression of a large number of transcripts involved in plant hormone signal-transduction, phenylpropanoid biosynthesis, carbon metabolism, and wound responses (Figs. 3–6 and Figs. 7–10). In grapevine, numerous gene expression changes related to wound responses, cell wall modification, hormone signaling, and secondary metabolism have been previously reported (Prodhomme et al., 2019). Our results again demonstrated that the metabolism and signal transduction pathways were activated during the graft healing process, indicating their essential regulatory roles in the graft healing formation.

The formation of callus tissue during graft healing is a necessary response to grafting. The lack of callus formation is a significant cause for grafting failure (Pina & Errea, 2005). Furthermore, callus differentiation into vascular tissues, which re-connects the xylem and phloem at the graft junction, is considered to be a crucial step for successful grafting (Flaishman et al., 2008). IAA, CTK, ABA, and GA are major hormones known to relate to callus formation, vascular bundle development, and reconnection (Bishopp et al., 2011; Mauriat & Moritz, 2009; Nieminen et al., 2008). In plants, auxin signaling is transmitted via transcriptional regulation of auxin early responsive gene families including AUX/IAA, Gretchen Hagen 3 (GH3), and Small Auxin Up RNA (SAUR) (Feng et al., 2015; Shen et al., 2014). Several auxin-responsive genes are thought to be regulated during the graft healing formation at the transcriptional level (Qiu et al., 2016; Zheng et al., 2009). In our study, most AUX/IAA and AUX1 genes were up-regulated at the VB stage in the melon scion (Figs. 11–14). The expression level was consistent with the auxin content, indicating that increasing auxin promotes vascular development during the graft healing process (Melnyk et al., 2015).

Similar to the results of a previous study in hickory (Mo et al., 2018), the auxin content in melon/squash graft junctions first decreased and then increased during the graft healing process. This finding suggests that auxin is transported from the scion to the rootstock and consequently triggers the differentiation of xylem cells (Caño Delgado, Lee & Demura, 2010) and plays an essential role in vascular development (Yin et al., 2012). Although it is reasonable to assume that the accumulation of auxin plays a vital role in melon/squash graft healing, how auxin promotes the vascular connection still requires further investigation.

Cytokinins are also known to participate in cell division and vascular differentiation (Bishopp et al., 2011; Hejátko et al., 2009). Therefore, we analyzed the genes related to cytokinin signaling during the graft healing process. In our RNA-seq data, the expression of two AHP genes (MELO3C015359.2 and MELO3C024439.2) was induced at the early stage of grafting (Fig. 12), indicating that these may be necessary for callus formation. Cytokinin signal-transduction is mediated by a two-component regulatory pathway that activates type-B ARR transcription factors (Pils & Heyl, 2009). Twenty-three functional responsive regulators (RRs) have been identified in Arabidopsis thaliana (El-Showk, Ruonala & Helariutta, 2013; Hwang, Sheen & Müller, 2012; To et al., 2007). However, only one A-type and two B-type RR genes were annotated in the transcriptome data presented here. Interestingly, the two B-type RR genes were up-regulated during the VB period, indicating their unknown functions explicitly relate to the formation of vascular bundles. Most of the gene expression in the melon scion was largely up-regulated during the early stage, indicating activation of cytokinin signaling during callus formation.

GAs play an important role in regulating plant growth by promoting cell proliferation and differentiation (Claeys, De Bodt & Inzé, 2014; Daviere & Achard, 2013; Yamaguchi, 2008). Recently, GAs have been reported to promote lignin formation in the cambium tissue (Mauriat & Moritz, 2009; Ragni et al., 2011), which may be crucial for the formation of vascular bundles during the graft healing process. This finding coincides with our observation of a significantly increased GA content at the VB stage. On the other hand, no significant difference in GA content was found when the IL stage was compared to the CA stage (Fig. 15D). Most unigenes encoding GA-signaling genes were induced at the VB stage, suggesting a role in vascular development. One TF (CmoCh09G007180) gene, however, was up-regulated at the IL stage indicating activation of GA-signaling during the graft healing process in oriental melon. Overall, these results showed that despite GAs likely involvement in the formation of graft healing, their functions could be not as crucial as auxin and cytokinin.

ABA plays essential regulatory roles during plant developmental processes including seed maturation, germination, and stomatal aperture (Finkelstein, 2013; Vishwakarma et al., 2017). A few studies have linked ABA to graft healing formation indirectly. However, our transcriptome data showed that the expression level of many genes encoding ABA-signaling components first increased and then decreased during the graft healing process. This suggests that ABA acts as a signaling molecule in response to wounding during the early stage and then is significantly down-regulated to inhibit the healing process. The ABA content also decreased at the CA stage and the VB stage. Previous studies have demonstrated that mutants with reduced ABA synthesis or signaling are more efficient at forming a wound-induced callus (Ikeuchi et al., 2017). Taken together, our findings suggested that ABA may inhibit graft healing and many genes encoding hormone signal transduction take part in the healing process, promoting melon graft formation.

Cellulose and lignin are two important biopolymers within the vascular bundle cell wall, and successful grafting in plants requires the development of a functional vascular system between the scion and rootstock. In a previously published grapevine graft study, cell wall precursors and lignin biosynthesis genes were also enriched in hetero-grafting compared to self-control due to differences in plant wounding and defense responses (Cookson et al., 2013). In the transcriptome analysis, we found 15 significantly differential genes encoding five enzymes involved in the lignin pathway. The expression levels of PAL genes in the squash rootstock and melon scion first decreased and then increased during the graft healing process. It has been previously reported that PAL genes can be regulated at the transcriptional level in response to different abiotic and biological stresses such as wounding, low temperatures, pathogen attack, and nutrient deficiency (Dixon et al., 2002; Olsen et al., 2008). Also, the up-regulation of PAL genes has been reported during graft healing development (Irisarri et al., 2016; Pina & Errea, 2008), suggesting that these genes could play an essential role in graft healing formation. Other genes related to COMT, 4CL, CCoAOMT, and CAD showed an increasing trend during the graft healing process, suggesting that these genes were also involved in graft healing and could function in promoting the development of vascular bundles and callus.

In conclusion, transcriptome analysis was performed to screen for DEGs involved in the melon/squash graft healing process. A total of 3,263 DEGs were identified from comparisons of the IL stage vs. the CA stage and the CA stage vs. the VB stage libraries. Based on our results, genes related to CTK-signaling transduction pathways in melon scion were up-regulated after grafting, maybe were essential to promote the callus formation. IAA and GA were then induced during the late stage. The activation signals may promote a successful grafting while ABA may be inhibitory for graft healing. In addition, genes related to lignin biosynthesis were up-regulated following grafting, potentially playing a vital role in graft healing formation. Taken together with findings from the transcriptome data, we demonstrated that the melon scion may take more of a leading role by participating more in metabolic activities and providing a more material basis for the graft healing process compared to the squash rootstock. The identification and analysis of key differentially expressed candidate genes will aid in the understanding of the complexity of hormone signaling and lignin biosynthesis during the graft healing process in plants.

Supplemental Information

Supplemental Information 1 PRISMA checklist

Click here for additional data file.

Supplemental Information 2 The lignin content of graft junction tissue at IL, CA, VB stage underlying the graft union development

Click here for additional data file.

Supplemental Information 3 Summary of transcriptome sequencing data generated from nine cDNA libraries

Click here for additional data file.

Supplemental Information 4 The classification of the enriched GO

Click here for additional data file.

Supplemental Information 5 The classification of the KEGGs

Click here for additional data file.

Supplemental Information 6 The expression level of plant hormone signaling transduction pathway-related genes

Click here for additional data file.

Supplemental Information 7 The expression level of lignin biosynthesis pathway-related genes

Click here for additional data file.

Supplemental Information 8 The specific primers used in fluorescence quantitative PCR detection

Click here for additional data file.

Supplemental Information 9 Raw data of lignin content during the graft healing process

Click here for additional data file.

Supplemental Information 10 Raw data of IAA, ABA, ZR and GA3 content during the graft healing process

Click here for additional data file.

Supplemental Information 11 Raw data of differential gene verification during the graft healing process

Click here for additional data file.

Supplemental Information 12 Accession number of sequencing data

Click here for additional data file.

Abbreviations

DAG day after grafting

IL isolated layer

CA callus

VB vascular bundle

qRT-PCR quantitative reverse transcription-polymerase chain reaction

RNA-seq transcriptome sequencing

DEGs differentially expressed genes

GC-MS gas-chromatography-mass spectrometry

HPLC high performance liquid chromatography

IAA indole-3-acetic acid

CTK cytokinin

ZR zeatin riboside

GA gibberellin

ABA abscisic acid

ELISA Enzyme-Linked Immunosorbent Assay

Additional Information and Declarations

Competing Interests

Author Contributions

Data Availability

The authors declare there are no competing interests.

Chuanqiang Xu and Ying Zhang conceived and designed the experiments, performed the experiments, analyzed the data, prepared figures and/or tables, authored or reviewed drafts of the paper, and approved the final draft.

Mingzhe Zhao conceived and designed the experiments, analyzed the data, prepared figures and/or tables, authored or reviewed drafts of the paper, and approved the final draft.

Yiling Liu and Tianlai Li conceived and designed the experiments, prepared figures and/or tables, and approved the final draft.

Xin Xu conceived and designed the experiments, performed the experiments, analyzed the data, prepared figures and/or tables, and approved the final draft.

The following information was supplied regarding data availability:

The raw data are available in the Supplemental Files. The sequencing data are available at NCBI SRA: PRJNA655799.

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
