# Peer review of "Transcriptomic analysis of melon/squash graft junction reveals molecular mechanisms potentially underlying the graft union development"

_PeerJ, doi:10.7717/peerj.12569_

## Round 0.1 · original submission · Major Revisions

The reviewers have made several very useful suggestions regarding the manuscript. In particular, more information regarding techniques (e.g. the method of grafting) and the suggested improvements to the figures are of particular importance. For each suggestion made by a reviewer, please either make the suggested alteration or explain why that particular suggestion was not followed.

Reviewer 1 ·

Basic reporting

The manuscript entitled “Transcriptomic analysis of melon/squash graft junction reveals molecular mechanisms underlying the healing development ”, by Xu et al. was reviewed.
The authors are interesting in the gene regulatory mechanism and involvements of graft formation of melon/squash, they performed transcriptome analysis to describe the morphological/transcriptional changes during graft formation in melon/squash.
In addition, the authors also analyzed endogenous levels of phytohormone and lignin of melon/squash graft, discussed about the involvement of auxin, GA, ABA, cytokinin and lignin on graft union .
The authors are discussing about relationship between morphological/physiological changes and gene regulatory network that occur during graft formation, this work contains new and important information for graft formation of plants.

Specific comments;
They used “IL, CA, and VB” as abbreviation for grafting stage, but these wording are not appropriate and difficult to understand. I propose to use “2DAG, 6DAG, 9DAG” or “Early, Mid, Late” or something similar in manuscript.

Table1) I propose to move this table to Supporting Information, because it is not informative.

Fig.1) These photographs should be revised. The orientation of each photos is different, and more explanation for results and abbreviation for symbols should be added in photo and figure legend. The authors also should indicate which is a scion or rootstock. In addition, the reviewer is wondering how authors determine callus formation and new vascular bundles. They described about it in text (line 176-191), but I cannot see such morphological changes from these images (B, C). The photograph should be replaced with better images and explain in more detail.

Fig.3,4,5) The text is too small, and the resolution is too low to read. These figures should be revised, higher resolution, split, or simpler, because it is difficult to see the data.

Fig. 5, 9) These figures need more information. It will look better if a gene name or deduced function were included in each graph.

I propose to cite these related-papers about plant grafting “Notaguchi et al., 2020; Science” and “Matsuoka et al., 2018; Plant Cell Physiol.”in the introduction or discussion, for the reader's better understanding.

Experimental design

Fig. 6) One section of the paper concerns about the relationship between phytohormone and graft formation. The authors showed that endogenous levels of phytohormone, but there are some concerns about experimental design. The reviewer is wondering why authors determine endogenous phytohormone levels using ELIZA, instead of LC-MS or LC-MS. Furthermore, they analyzed levels of auxin, GA, ABA, cytokinin. Although GA3 is one of major bioactive GA, but generally, not detected in wildtype of higher plant. The reviewer is also wondering why such amount of GA3 was detected in this work. In addition, zeatin riboside is not bioactive form in higher plants. Please explain why authors used this method and analyze these hormones, and I strongly recommended that authors should confirm experimental condition, data analysis, and perform quantitative analysis of endogenous bioactive GA and cytokinins. Moreover, there is no control experiments. Analysis of phytohormone using non-grafted plants should be done.

They mentioned hormone content during the graft process and discussed about the relationship with gene expression and morphological changes. There were statistically significant changes, but I am concerned about whether these are biologically meaningful changes, except for ABA. Explanation needs more detail.

Validity of the findings

While the results are intriguing and may potentially lead to greater understanding of molecular mechanisms underlying melon/squash graft formation, no direct connection between gene function and graft union formation has been shown and there are some concerns about the interpretations of the results and illustration /description of figures.

Additional comments

The text is clearly described, but the illustration, description and legends of each figures were not clear, so an effort should be made by authors to clarity of the manuscript.

Reviewer 2 ·

Basic reporting

no comment

Experimental design

see my comments in general comments.

Validity of the findings

no comment

Additional comments

The study is nice and adds to the existing information. My minor comments are:

Line #111: What are the control conditions in sample collection?

Line #161: Why did you selected only IAA, ZR, GA3, ABA?

Line # 175: If you did microscopy continuously then where is the remaining data?

Line # 176: The graft healing formation was divided into three recognizable developmental stages, so what was the basis of division of the developmental stages?

Reviewer 3 ·

Basic reporting

Xu et al present a transcriptomic study of the graft interface of melon/squash grafts at 2, 6 and 9 days after grafting, this study is complemented by some microscope images and hormone analysis. The manuscript is well-written and the results are interesting. However, some references need to be modified and more details are required in the methods section.

In the abstract, the authors need to define what DEGs, IL, CA and VB are.

Experimental design

Methods
More details of the grafting technique should be given, what stage were the grafts performed? What were the growth conditions? What was the grafting success? Etc.
More details are required in the methods section such as how was the graft interface sampled? For example, 2 mm above and below the graft interface? Where the same samples used for the RNAseq, lignin and hormone analysis?
For the RNAseq analysis, it is interesting and novel that the transcripts were assigned separately to the scion and rootstock, but more details of the methods need to be given, such as, how were transcripts assigned to the melon scion or squash rootstock? What reference genomes were used? It seems that more reads mapped to the stock than the scion, is this due to the sample? Did it contain more rootstock than scion tissue? What percentage of transcripts was ambiguous in their assignment? What did the authors do with the transcripts that could not be assigned to either the scion or rootstock genotype?
The different gene expression patterns in the scion and rootstock should be discussed in relation to the study of Melnyk et al 2018.
If the RNAseq, lignin and hormone analysis samples are the same, it could be worth doing a WGCNA analysis of the data to find transcripts strongly correlated to hormone/lignin concentration.

Validity of the findings

The validity of the findings is generally OK, but a few changes are required.
Title: change to “Transcriptomic analysis of melon/squash graft junction reveals molecular mechanisms POTENTIALLY underlying the GRAFT UNION development”
Lines 282-285, lines 323-324, lines 388-389, lines 413-415: confirmatory statements should be made more modest with works like suggests/may/could etc.

Additional comments

The different gene expression patterns in the scion and rootstock should be discussed in relation to the study of Melnyk et al 2018.

Figure 1: The images should be better annotated, the graft interface, scion and rootstock should be labelled, etc. It might be good to have better images of the beginning of callus cell formation and vascular bundles and some idea of where in the graft interface the images were taken. This would also help the reader see what you want them to see in the images.
Figure 8: Error bars missing
The details of the data repository (NCBI) should be given in the methods section.
What is CTK signaling? (line 282)
What is the reference for this statement “In addition, a blockage in auxin transport due to vascular damage lead to decreased expression of GH3 genes during the early stage, suggesting a response for wounding during graft healing.” (Line 348-350)
Reid and Ross 2011 is a commentary on a paper, it is preferable to cite original research.
Line 399: should be Cookson et al 2014

---

## Round 0.2 · Minor Revisions

The reviewers and I appreciate the improvements you have made to your manuscript. However, prior to publication, the manuscript would benefit from some additional editing. In particular, please address Reviewer 1's comments about the designations used for the grafting stages and respond to the comment about whether GA3 could have been mis-identified and might actually represent a different gibberellin. Also, it is essential that the figures be readable, so please edit the figures as suggested by Reviewer 3, in particular making sure that the font is large enough to be readable.

Reviewer 1 ·

Basic reporting

no comment

Experimental design

I was still confused with the words of "“IL, CA, and VB” as abbreviation for grafting stage, in this figure and the text.
The reviewer understood the “IL, CA, and VB” as a key events in the process of grafting healing, but these wording are not appropriate for abbreviation of grafting stage this way.
The authors use the abbreviation of “IL, CA, and VB” in two means, name of structure and grafting stage. It would be clearer to give each abbreviation used with its explanation.
Again, I propose to use “2DAG, 6DAG, 9DAG” or “Early, Mid, Late” or, at least, “IL stage, CA stage, and VB stage” for grafting stage or something similar in manuscript.


The authors described the reason of using ELIZA, instead of LC-MS or LC-MS, to determine endogenous phytohormone levels, but did not explain why such amount of GA3 was detected in this work. The authors may have misunderstood GA3 for another GA or may be an artifact of experimental design.
As pointed out previously, GA3 is one of major bioactive GA, but generally, not endogenous GA, and not detected in wildtype of higher plant, including melon/squash Solanaceae plants, these results are not justified. If GA3 was detected, the authors should give some references and/or convincing evidence.

Validity of the findings

no comment

Additional comments

The revised manuscript entitled "Transcriptomic Analysis of Melon/Squash Graft Junction Reveals Molecular Mechanisms Potentially Underlying the Graft Union Development, by Xu et al. " was reviewed. I feel that this manuscript has been improved satisfactorily since some sentences and figures were added and improved, but there are still some points to be revised.

Reviewer 3 ·

Basic reporting

The basic reporting of this manuscript is correct.

Experimental design

The experimental design is OK, but care should be taken in the interpretation of the data: as only the graft interface is compared over time, the authors cannot separate changes due to development and those due to graft union formation. Ideally scion and rootstock tissues should have been studied at the same time to identify graft union formation specific genes.

Validity of the findings

The findings seem to be OK, comments have been added to the word document of the manuscript.

Additional comments

I have some comments about the figures:
Figure 1: This figure could be improved, the different cell types are not particularly clear.
Figure 3: This figure is not very readable, the data may be better presented in a Table and maybe focus only on the over-represented categories and not the under-represented categories (like reproduction, etc).
Figure 4: the text is too small and unreadable. It is very surprising to find differences in circadian rhythm, were the plants all harvested at the same point in the photoperiod? Or is it due to changes in carbon availability? If they were not harvested as the same point in the diurnal cycle this should be added to the methods section
Figure 5: the text is too small and should be made simpler.

Annotated reviews are not available for download in order to protect the identity of reviewers who chose to remain anonymous.

---

## Round 0.3 · Minor Revisions

The improvements made to the manuscript are appreciated. However, unfortunately the font size does not appear to have been increased in most of the figures. It is still too small to read, particularly in figures 2, 3, 4, 5, 6 and 8.

---

## Round 0.4 · Minor Revisions

Thank you for your improvements to the figures. They are now much easier to read.

Before it can be accepted, please attend to the following comments from the Section Editor:

> Some minor changes needed to figures. Figs 7-10 "Rich Factor" is not standard. How about "Enrichment Factor" or perhaps even better "Fold Enrichment".
>
> Figures 11-14 "Log" of what? cpm? fpkm? raw counts?

---

## Round 0.5 · accepted · Accept

Thanks for revising your manuscript.